# Predefined-Time Adaptive Neural Tracking Control for a Single Link Manipulator with an Event-Triggered Mechanism

**DOI:** 10.3390/s24144573

**Published:** 2024-07-15

**Authors:** Yikai Wang, Yuan Sun, Yueyuan Zhang, Jun Huang

**Affiliations:** School of Mechanical and Electrical Engineering, Soochow University, Suzhou 215137, China; 2129402087@stu.suda.edu.cn (Y.W.); cauchyhot@163.com (J.H.)

**Keywords:** adaptive control, event-triggered control, nonlinear systems, predefined-time control

## Abstract

This paper introduces an adaptive trajectory-tracking control method for uncertain nonlinear systems, leveraging a time-varying threshold event-triggered mechanism to achieve predefined-time tracking. Compared to conventional time-triggering approaches, the employment of a time-varying threshold event-triggered mechanism significantly curtails communication resource wastage without compromising the system’s performance. Furthermore, a novel adaptive control algorithm with predefined timing is introduced. This method guarantees that tracking errors converge to within a small vicinity of the origin within a predefined timeframe, ensuring all signals in the closed-loop system remain bounded. Moreover, by adjusting a controller-related parameter, we can predefine the upper bound of the convergence time. Finally, the efficacy of the control scheme is corroborated by simulation results obtained from a nonlinear manipulator system.

## 1. Introduction

Robotic manipulators, which are complex mechanical systems that mimic human arm movements, exhibit high nonlinearity, strong coupling, and superior controllability. Their applications are significant across various fields, including industrial manufacturing, medical treatment, military, semiconductor production, and space exploration [1,2]. Consequently, the challenges of controlling robotic manipulators have garnered considerable interest and research from numerous scholars [3,4].

As control technology has advanced, a multitude of control algorithms have been successfully implemented in robotic manipulator control, yielding significant achievements. For instance, PID control stands as one of the most extensively deployed control algorithms [5,6]. The a PID controller adjusts robotic manipulators’ movement by calculating the control inputs based on the error between the desired and the actual positions, along with the error’s integral and derivative. However, its complex parameter tuning process during control complicates the controller design and debugging. Additionally, in the presence of disturbances or when a rapid response is required, the PID control may fail to adjust control inputs in a timely manner, resulting in system response delays or overshooting. Consequently, developing more robust control algorithms becomes imperative.

To enhance control performance, numerous control algorithms prioritize addressing the uncertainties inherent in robotic manipulator systems [7]. Adaptive control offers semi-global or global stability for nonlinear systems with parameter uncertainties [8,9]. However, as demands for higher performance escalate and system uncertainties intensify, applying adaptive control to increasingly complex systems presents challenges, including the complexity of adaptive controller structures and the inability of adaptive control systems to guarantee steady-state errors against disturbances. The emergence of adaptive neural network control in recent years has offered fresh perspectives on surmounting the constraints of conventional adaptive control approaches [10]. Leveraging neural networks’ robust capabilities for approximating nonlinear functions, adaptive neural network control strategies can precisely emulate the dynamic behaviors of complex systems, enhancing control accuracy and adaptability. Consequently, adaptive control strategies utilizing neural network approximations have been extensively applied in nonlinear systems [11,12]. An adaptive impedance controller utilizing a radial-basis-function neural network (RBFNN) is presented for robotic manipulator systems characterized by uncertainties and input saturation, with the goal of facilitating precise control, in [11]. The authors of [12] leverage RBFNN to approximate and compensate for uncertainties in the model, introducing an adaptive robust control algorithm based on RBFNN. This algorithm utilizes robust control to significantly diminish approximation errors, culminating in successful trajectory tracking for robotic manipulators. On the other hand, given that robotic manipulator systems can be suitably transformed into strict feedback systems, the adaptive backstepping control approach emerges as superior. It designs control laws in an iterative manner, skillfully managing the strict feedback configuration and effectively addressing the system’s nonlinearities and uncertainties, thereby ensuring the stability of the entire closed-loop system [13]. Consequently, the adaptive backstepping control method is employed in complex nonlinear systems, such as robotic manipulators and wheeled mobile robots, to achieve precise position control and trajectory tracking [14,15]. Leveraging backstepping technology introduces an adaptive controller based on the least squares support vector machine, aimed at precise robotic manipulator positioning. This method exhibits enhanced robustness, reduced tracking error, and improved response speed relative to traditional PD control. The authors explored a backstepping control approach for robotic manipulators with flexible joints in [14,15]. The former approach eliminates the need for measuring joint angular acceleration, while the latter addresses system vibrations. Both strategies facilitate effective position control.

The studies mentioned above have yielded positive outcomes in enhancing the performance and efficacy of robotic manipulators’ trajectory tracking. However, the transmission of control signals typically operates at fixed time intervals, necessitating shorter sampling periods for stability and effectiveness, but excessively high sampling rates lead to unnecessarily frequent updates following system stabilization. This often results in the wastage of communication resources. The event-triggered mechanism has effectively addressed the aforementioned issue [16]. The core principle behind event-triggered mechanisms is that communication data for measurement signals are transmitted only upon satisfying the design criteria of the event-driven strategy, thereby conserving communication resources while maintaining system control performance. To date, event-triggered control, as a strategy significantly reducing signal communication, has garnered extensive research [17,18,19]. The study in [17], focusing on robotic manipulator systems, explores the interplay between sliding mode controllers and event-triggered mechanisms. This approach enables precise trajectory tracking with fewer communication and signal updates, effectively eliminating Zeno phenomena. The authors of [18] introduced a periodic event-triggered control design tailored for nonlinear robotic manipulator systems. This design notably decreases the controller update frequency by minimizing the need for continuous state measurements, thereby conserving computational resources. The authors developed various event-triggered mechanisms to enhance the flexibility and energy efficiency of the system in [20,21]. However, the uncertainty of robotic manipulator models, external disturbances, and the imperative to avert Zeno phenomena during control present significant challenges in applying event-triggered strategies to robotic manipulator systems.

Importantly, the control methods discussed primarily focus on the system’s asymptotic stability or uniform ultimate boundedness. This suggests that achieving system stability within a finite timeframe or reaching a stable control objective within a specified period might not be feasible. In scenarios requiring precise timing for robotic manipulator operations, traditional incremental control algorithms fall short of meeting these practical needs. To overcome this challenge, robotic manipulator control has incorporated finite-time [22,23] and fixed-time [24,25] control theory. However, the convergence efficacy of finite-time control significantly depends on the initial condition of the system, often making it challenging to attain satisfactory convergence in practical scenarios [26]. The adoption of fixed-time control effectively addresses this issue by decoupling convergence time from initial conditions. However, fixed-time control does not facilitate a direct relationship between convergence time and controller parameters, potentially complicating the design and adjustment of parameters to align with real-world system requirements. Recently, a predefined-time control strategy has successfully addressed the limitations associated with finite-time and fixed-time control [27,28,29,30]. This approach is able to predetermine the upper bound for settling time, ensuring a high-performance temporal response from the system. Consequently, this emerging technology possesses significant developmental potential in the context of robot applications [31,32]. The authors of [31] concentrated on the issue of predefined-time stability in the posture control of multi-segment cable-driven spatial continuous robotic systems, introducing an effective solution via an adaptive controller employing the non-singular terminal sliding mode. Reference [32] introduces a novel adaptive predefined-time tracking control strategy for robotic manipulators facing input saturation challenges. By incorporating an auxiliary dynamic system, this approach adeptly navigates input constraints, securing the posture convergence of the manipulator within a predetermined timeframe. Consequently, based on the aforementioned analysis, investigating adaptive predefined-time stability for robotic manipulator systems with uncertainties is of substantial practical significance.

Analyzing the above literature reveals that in robotic manipulator systems with inherent uncertainty, we face the critical challenge of reducing communication resource usage while maintaining system control performance. Thus, incorporating an event-triggered mechanism holds significant value but presents certain challenges. Moreover, designing an attitude controller that meets convergence time requirements is crucial for the practical use of robotic arms in real-world applications. However, to the best of our knowledge, research on adaptive neural network-based event-triggered mechanisms and predefined-time robotic arm control remains underexplored. These issues warrant further investigation.

Drawing inspiration from prior research, this study focuses on an adaptive predefined-time trajectory tracking control problem with an event-triggered mechanism for robotic manipulator systems. The devised control law guarantees tracking error convergence to a small vicinity of the origin within a predefined period, while ensuring all closed-loop signals stay bounded within the same timeframe. The key contributions and highlights of this study are summarized as follows:(1)Diverging from finite-time [22,23] and fixed-time [24,25] control theories, the controller introduced in this study guarantees tracking error convergence to a designated area within a predefined time. Adjusting a single control parameter allows for the precise setting of the settling time’s upper bound, independent of initial conditions, aligning with practical engineering demands for system convergence time and precision.(2)Departing from the conventional adaptive predefined-time controls in [27,28,29], this work introduces a novel adaptive law, where a unique adaptive parameter is determined by the norm of neural network weights, thereby significantly simplifying computational complexity.(3)Contrasting with traditional event-triggered control approaches in [11,12,13], this study integrates a time-varying threshold event-triggered mechanism, effectively conserving communication resources without compromising system control efficacy.
The remainder of this manuscript is structured as follows: Section 2.1, Section 2.2 and Section 2.3 outline the mathematical model and preliminaries. Section 2.4 demonstrates the controller design and stability analysis. Section 2.5 proves the feasibility of the proposed method. Section 3 presents simulation results that validate the effectiveness of the designed controller. Section 4 provides a detailed analysis and discussion of the simulation results. Section 5 provides a comprehensive summary of the document.

Notations: In this paper, $∥\cdot ∥$ denotes the Euclidean two-dimensional norm. We define ${\mathrm{sig}}^{\gamma }\left(a\right)={\left|a\right|}^{\gamma }\mathrm{sgn}\left(a\right)$, with a∈R and a constant γ∈R, where sgn(·) serves as the sign function.

## 2. Methods

### 2.1. Model of a Robotic Manipulator System

Consider a robotic manipulator system, with its dynamical model describable as follows:(1)$$I\ddot{q}+c_{0}\dot{q}+mgl\cos\left(q\right)=\tau +d$$
where q∈R represents the arm angle, $\dot{q}\in R$ denotes angular velocity, $\ddot{q}\in R$ is angular acceleration, τ∈R refers to input torque, $I=\frac{4}{3}ml^{2}$ denotes moment of inertia, *m* shows the arm’s weight, *l* is the arm’s length, $c_{0}$ denotes the viscous friction coefficient, *d* represents continuous bounded input disturbances, and *g* stands for the gravitational acceleration.

Let $x_{1}=q$ and $x_{2}=\dot{q}$; then, system (Equation 1) can be reformulated as follows:(2)$$\left\{\begin{array}{c} {\dot{x}}_{1}=x_{2}\\ {\dot{x}}_{2}=f+g_{1}u\\ y=x_{1} \end{array}\right.$$
where u=τ denotes the system input, and $y=x_{1}$ represents the system output, with $f=-\frac{1}{I}\left[c_{0}{\dot{x}}_{1}+mgl\cos\left(x_{1}\right)-d\right]$ being an uncertain nonlinear function and $g_{1}=\frac{1}{I}$ a known constant.

This paper aims to devise an adaptive predefined-time control scheme utilizing an event-triggered mechanism for nonlinear robotic manipulator systems, ensuring that

(1)The position of the robotic manipulator joint $x_{1}$ can precisely follow the reference angle $y_{d}$, with the tracking error rigorously confined within a compact set;(2)All signals within the closed-loop robotic manipulator system remain bounded within the predefined timeframe;(3)According to the proposed controller, it can significantly reduce the consumption of communication resources without compromising control precision.

### 2.2. Preparatory Work

**Lemma** **1**(See [19])**.**
(3)$$0\leq \left|\xi \right|-\xi \tanh\left(\frac{\xi }{\rho }\right)\leq \delta \rho$$*where ξ∈R, δ=0.2785 and ρ>0 are constants.*

**Lemma** **2**(See [29])**.**
*For conditions $x_{i}\geq 0\left(i=1,2,3\ldots n\right)$ and γ>0, the following inequalities are valid.*
(4)$$\left\{\begin{array}{c} \sum_{i=1}^{n}x_{i}^{\gamma }\geq {\left(\sum_{i=1}^{n}x_{i}\right)}^{\gamma },0<\gamma <1\\ \sum_{i=1}^{n}x_{i}^{\gamma }\geq n^{1-\gamma }{\left(\sum_{i=1}^{n}x_{i}\right)}^{\gamma },\gamma >1 \end{array}\right.$$

**Lemma** **3**(See [30])**.**
(5)$$x{\left(y-x\right)}^{v}\leq \frac{v}{1+v}\left(y^{1+v}-x^{1+v}\right)$$*where y≥x and v>1.*

**Lemma** **4**(See [30])**.**
*For system $\dot{x}=f\left(t,x\right)$, we define a continuous function V(x) and establish parameters 0<β<1, $T_{c}>0$, and 0<σ<∞ to fulfill the condition*
(6)$$\dot{V}\left(x\right)\leq -\frac{\pi }{\beta T_{c}}\left(V^{1-\frac{\beta }{2}}+V^{1+\frac{\beta }{2}}\right)+\sigma$$
*consequently, the trajectory of $\dot{x}=f\left(t,x\right)$ is practically predefined-time stable (PPTS), and its convergence zone is*
(7)$$\left\{\lim_{t\to T_{P}}x|V\leq \min\left\{{\left(\frac{2\beta T_{c}\sigma }{\pi }\right)}^{\frac{2}{2-\beta }},{\left(\frac{2\beta T_{c}\sigma }{\pi }\right)}^{\frac{2}{2+\beta }}\right\}\right\}$$*where $T_{P}$ represents the settling time and meets condition $T_{P}<T_{\max}=\sqrt{2}T_{c}$, with $T_{\max}$ as its upper bound.*

**Remark** **1.**
*$T_{c}$ is a predefined time parameter. According to Lemma 4, if a Lyapunov function exists in the form of Equation (Equation 6), then the system is predefined-time stable. In this scenario, the upper bound of the system’s settling time is $\sqrt{2}T_{c}$. Clearly, this upper time bound is entirely artificially set and independent of system parameters and initial conditions. By adjusting $T_{c}$, we can directly alter the upper bound of the system’s settling time, indicating that the error-convergence time can be preset by parameter adjustment.*


**Lemma** **5**(See [33])**.**
(8)$$0\leq \left|z\right|-\frac{z^{2}}{\sqrt{z^{2}+\kappa^{2}}}<\kappa$$*where κ>0 is a constant, with z∈R.*

**Lemma** **6**(See [34])**.**
(9)$$\dot{\hat{\theta }}=h_{1}\omega \left(t\right)-h_{2}\hat{\theta }\left(t\right)-h_{3}{\hat{\theta }}^{\mu_{1}}\left(t\right)$$*where $h_{1},h_{2},h_{3}>0,\mu_{1}>1$ are constants, and ω(t) is a non-negative function. Consequently, if $\hat{\theta }\left(0\right)\geq 0$, then for any t≥0, $\hat{\theta }\left(t\right)\geq 0$ holds.*

**Lemma** **7**(See [35])**.**
(10)$$\begin{array}{cc} \gamma_{1}^{\pi_{1}}\gamma_{2}^{\pi_{2}}\gamma_{3}\leq & \pi_{3}\gamma_{1}^{\pi_{1}+\pi_{2}}+\frac{\pi_{2}}{\pi_{1}+\pi_{2}}\\ & \times {\left[\frac{\pi_{1}}{\pi_{3}\left(\pi_{1}+\pi_{2}\right)}\right]}^{\frac{\pi_{1}}{\pi_{2}}}\gamma_{2}^{\pi_{1}+\pi_{2}}\gamma_{3}^{\frac{\pi_{1}+\pi_{2}}{\pi_{2}}} \end{array}$$*where $\pi_{1}>0,\pi_{2}>0,\pi_{3}>0,\gamma_{1}\geq 0,\gamma_{2}\geq 0,\gamma_{3}\geq 0$.*

**Assumption** **1.**
*The ideal reference signal $y_{d}$, its derivative ${\dot{y}}_{d}$, and its second derivative ${\ddot{y}}_{d}$ are continuous and bounded.*


### 2.3. Radial Basis Neural Network (RBFNN)

The RBFNN, a prevalent three-layer feedforward architecture, comprises an input layer, a hidden layer, and an output layer. The input layer, formed by source nodes, receives raw data. The hidden layer applies radial basis functions for nonlinear data transformation, while the output layer linearly weights and sums the hidden layer’s outputs to produce the outcome. Its network structure is shown in Figure 1.

In this paper, an RBFNN defined over the compact set $\Omega_{X}\in R^{n}$ is employed to approximate the uncertain nonlinear function $\varphi \left(X\right):R^{n}\to R$. The specific formulation is as follows:(11)$$\varphi \left(X\right)=W^{T}S\left(X\right)+\iota \left(X\right)$$
where $X={\left[x_{1}\cdots x_{n}\right]}^{T}$ represents the input vector, with $n\in N^{*}$. The term ι(X) signifies the neural network’s intrinsic approximation error, and $\left|\iota (X)\right|\leq \varepsilon$, given that ε>0. The vector $S\left(X\right)={\left[s_{1}\left(X\right),\cdots s_{l}\left(X\right)\right]}^{T}$ represents the radial basis functions, and $l\in N^{*}$ denotes the number of nodes in the neural network’s hidden layer.

In this study, the Gaussian function is utilized as the radial basis function; thus,
(12)$$s_{i}\left(X\right)=\exp\left(-\frac{{∥X-\mu_{i}∥}^{2}}{\eta^{2}}\right)$$
where $\mu =\left[\begin{array}{ccc} \mu_{11} & \cdots & \mu_{1l}\\ \vdots & \ddots & \vdots \\ \mu_{n1} & \cdots & \mu_{nl} \end{array}\right]$ denotes the center of the domain for the Gaussian function, and $\mu_{i}$ represents the *i*th column component of μ, where i=1⋯l. The constant η>0 denotes the width of the Gaussian function. The ideal weight vector $W={\left[w_{1}\cdots w_{l}\right]}^{T}$ is defined as follows:(13)$$W=\arg\min_{\hat{W}\in R^{l}}\left\{\underset{X\in \Omega_{X}}{\sup}\left|\varphi \left(X\right)-{\hat{W}}^{T}S\left(X\right)\right|\right\}$$
where $\hat{W}$ represents the estimated value of *W*.

### 2.4. Controller Design

In the previous section, we introduced the system model, mathematical lemmas, and RBFNN, which are essential for the design of the controller algorithm. Next, in this section, we will use the backstepping technique to develop an adaptive predefined-time control strategy based on an event-triggered mechanism.

To clearly and intuitively present the adaptive predefined-time tracking control algorithm under the event-triggered mechanism proposed in this section, the following control scheme block diagram is provided in Figure 2.

#### 2.4.1. Adaptive Predefined-Time RBFNN Control Design

In this section, we employ backstepping techniques to design a predefined-time virtual controller and construct the corresponding Lyapunov functions. Additionally, RBFNN is used to estimate the uncertainties in the robotic manipulator system, and suitable adaptive laws are designed accordingly.

Initially, considering system (Equation 2), the coordinate transformation is defined as follows:(14)$$\left\{\begin{array}{c} z_{1}=x_{1}-y_{d}\\ z_{2}=x_{2}-\alpha_{1} \end{array}\right.$$
where $y_{d}$ represents the ideal tracking signal, and $\alpha_{1}$ denotes the first virtual control term, as specified by the subsequent equation:(15)$$\begin{array}{cc} \alpha_{1} & =-\frac{z_{1}{\overset{ˇ}{\alpha_{1}}}^{2}}{\sqrt[{\;}]{z_{1}^{2}{\overset{ˇ}{\alpha_{1}}}^{2}+\varepsilon_{1}^{2}}}+{\dot{y}}_{d} \end{array}$$
(16)$$\overset{ˇ}{\alpha_{1}}=\frac{\pi }{\beta T_{c}}\left[2^{\beta }{\left(\frac{1}{2}\right)}^{1+\frac{\beta }{2}}{\mathrm{sig}}^{1+\beta }\left(z_{1}\right)+{\left(\frac{1}{2}\right)}^{1-\frac{\beta }{2}}{\mathrm{sig}}^{1-\beta }\left(z_{1}\right)\right]$$
where $\varepsilon_{1}$, 0<β<1 are positive constants, and $T_{c}>0$ represents a predefined constant.

**Remark** **2.**
*In backstepping control approaches, ensuring the boundedness of the virtual controller and its first derivative is imperative. In traditional control strategies, the existence of ${sig}^{1-\beta }\left(z_{1}\right)$, with 0<β<1, leads to the first derivative of the virtual controller approaching infinity as $z_{1}$ approaches 0. Consequently, singularity issues arise in the first derivative of the virtual controller. Consequently, this study designs a virtual controller (Equation 15), addressing the issue. At this stage, ${\dot{\alpha }}_{1}$ consistently exhibits the property $\lim_{z_{1}\to 0}z_{1}{\left|z_{1}\right|}^{-\beta }=0$. This demonstrates the boundedness of ${\dot{\alpha }}_{1}$. Thus, the singularity issue has been effectively addressed.*


Derived from Equations (Equation 2) and (Equation 14),
(17)$$\begin{array}{cc} {\dot{z}}_{1} & ={\dot{x}}_{1}-{\dot{y}}_{d}\\ \; & =z_{2}+\alpha_{1}-{\dot{y}}_{d} \end{array}$$

We design the following Lyapunov function:(18)$$V_{1}=\frac{1}{2}z_{1}^{2}$$

By considering Equations (Equation 15) and (Equation 17), the first derivative of $V_{1}$ yields
(19)$$\begin{array}{cc} {\dot{V}}_{1} & =z_{1}\left(z_{2}+\alpha_{1}-{\dot{y}}_{d}\right)\\ \; & =z_{1}z_{2}-\frac{z_{1}^{2}{\overset{ˇ}{\alpha_{1}}}^{2}}{\sqrt[{\;}]{z_{1}^{2}{\overset{ˇ}{\alpha_{1}}}^{2}+\varepsilon_{1}^{2}}} \end{array}$$

Integrating Lemma 5 with Equation (Equation 16), we derive
(20)$$\begin{array}{cc} -\frac{z_{1}^{2}{\overset{ˇ}{\alpha_{1}}}^{2}}{\sqrt[{\;}]{z_{1}^{2}{\overset{ˇ}{\alpha_{1}}}^{2}+\varepsilon_{1}^{2}}}\; & <\varepsilon_{1}-\left|z_{1}\overset{ˇ}{\alpha_{1}}\right|\\ \; & <\varepsilon_{1}-\frac{\pi }{\beta T_{c}}{\left(\frac{z_{1}^{2}}{2}\right)}^{1-\frac{\beta }{2}}-\frac{\pi }{\beta T_{c}}2^{\beta }{\left(\frac{z_{1}^{2}}{2}\right)}^{1+\frac{\beta }{2}} \end{array}$$

Substituting (Equation 20) into (Equation 19), we obtain:(21)$${\dot{V}}_{1}<-\frac{\pi }{\beta T_{c}}{\left(\frac{z_{1}^{2}}{2}\right)}^{1-\frac{\beta }{2}}-\frac{\pi }{\beta T_{c}}2^{\beta }{\left(\frac{z_{1}^{2}}{2}\right)}^{1+\frac{\beta }{2}}+\varepsilon_{1}+z_{1}z_{2}$$

According to (Equation 2) and (Equation 14), we derive
(22)$$\begin{array}{c} {\dot{z}}_{2}={\dot{x}}_{2}-{\dot{\alpha }}_{1}=f-{\dot{\alpha }}_{1}+g_{1}u \end{array}$$

The Lyapunov function selected is as follows:(23)$$V_{2}=V_{1}+\frac{1}{2}z_{2}^{2}+\frac{{\tilde{\theta }}^{2}}{2r}$$
where $\tilde{\theta }=\theta -\hat{\theta }$ denotes the adaptive parameter error, and r>0 represents a constant.

The first derivative of $V_{2}$ yields
(24)$$\begin{array}{c} {\dot{V}}_{2}={\dot{V}}_{1}+z_{2}{\dot{z}}_{2}-\frac{1}{r}\tilde{\theta }\dot{\hat{\theta }} \end{array}$$

Substituting (Equation 21) and (Equation 22) into (Equation 24), we obtain
(25)$$\begin{array}{cc} {\dot{V}}_{2}< & -\frac{\pi }{\beta T_{c}}{\left(\frac{z_{1}^{2}}{2}\right)}^{1-\frac{\beta }{2}}-\frac{\pi }{\beta T_{c}}2^{\beta }{\left(\frac{z_{1}^{2}}{2}\right)}^{1+\frac{\beta }{2}}+z_{1}z_{2}+z_{2}{\dot{z}}_{2}\\ & -\frac{1}{r}\tilde{\theta }\dot{\hat{\theta }}+\varepsilon_{1}\\ < & -\frac{\pi }{\beta T_{c}}{\left(\frac{z_{1}^{2}}{2}\right)}^{1-\frac{\beta }{2}}-\frac{\pi }{\beta T_{c}}2^{\beta }{\left(\frac{z_{1}^{2}}{2}\right)}^{1+\frac{\beta }{2}}+z_{2}\varphi +z_{2}g_{1}u\\ & -\frac{1}{2}z_{2}^{2}-\frac{1}{r}\tilde{\theta }\dot{\hat{\theta }}+\varepsilon_{1} \end{array}$$
where $\varphi =f-{\dot{\alpha }}_{1}+z_{1}+\frac{1}{2}z_{2}$ represents an uncertain nonlinear function.

Using Lemma 1, we obtain
(26)$$\begin{array}{c} \left|z_{2}\right|∥S(X)∥\leq \delta \rho +z_{2}|∥S(X)∥\tanh\left(\frac{z_{2}|∥S(X)∥}{\rho }\right) \end{array}$$
where δ=0.2785, with ρ>0 representing a constant. Furthermore, $X={\left[x_{1},x_{2},y_{d},{\dot{y}}_{d}\right]}^{T}$ serves as the input to the RBFNN, and S(X) denotes the output vector from the hidden layer nodes of the RBFNN.

Utilizing RBFNN to approximate φ yields
(27)$$\begin{array}{c} z_{2}\varphi =z_{2}W^{T}S\left(X\right)+z_{2}\iota \left(X\right),\left|\iota (X)\right|\leq \varepsilon ,\varepsilon >0 \end{array}$$

Substituting (Equation 26) into (Equation 27) yields
(28)$$\begin{array}{cc} z_{2}\varphi \leq & \left|z_{2}\right|∥W^{T}∥∥S(X)∥+\frac{1}{2}z_{2}^{2}+\frac{1}{2}\varepsilon^{2}\\ \leq & ∥W^{T}∥\left[\delta \rho +z_{2}∥S(X)∥\tanh\left(\frac{z_{2}∥S(X)∥}{\rho }\right)\right]\\ \; & +\frac{1}{2}z_{2}^{2}+\frac{1}{2}\varepsilon^{2}\\ \leq & \delta \theta g_{1}\rho +z_{2}\theta g_{1}∥S(X)∥\tanh\left(\frac{z_{2}∥S(X)∥}{\rho }\right)+\frac{1}{2}z_{2}^{2}\\ & +\frac{1}{2}\varepsilon^{2} \end{array}$$
where $\theta =\frac{∥W^{T}∥}{g_{1}}$.

Substituting (Equation 28) into (Equation 25) leads to
(29)$$\begin{array}{cc} {\dot{V}}_{2}< & -\frac{\pi }{\beta T_{c}}{\left(\frac{z_{1}^{2}}{2}\right)}^{1-\frac{\beta }{2}}-\frac{\pi }{\beta T_{c}}2^{\beta }{\left(\frac{z_{1}^{2}}{2}\right)}^{1+\frac{\beta }{2}}+\delta \theta g_{1}\rho \\ & +z_{2}\theta g_{1}∥S(X)∥\tanh\left(\frac{z_{2}∥S(X)∥}{\rho }\right)+\frac{1}{2}\varepsilon^{2}\\ & +z_{2}g_{1}u-\frac{1}{r}\tilde{\theta }\dot{\hat{\theta }}+\varepsilon_{1} \end{array}$$

Furthermore, we can design the second virtual controller $\alpha_{2}$ and the predefined-time adaptive law $\dot{\hat{\theta }}$ as follows:(30)$$\begin{array}{cc} \alpha_{2} & =\frac{-z_{2}{\overset{ˇ}{\alpha_{2}}}^{2}}{g_{1}\sqrt{z_{2}^{2}{\overset{ˇ}{\alpha_{2}}}^{2}+\varepsilon_{2}^{2}}} \end{array}$$
(31)$$\begin{array}{cc} \overset{ˇ}{\alpha_{2}}= & \frac{\pi }{\beta T_{c}}\left[2^{\beta }{\left(\frac{1}{2}\right)}^{1+\frac{\beta }{2}}{\mathrm{sig}}^{1+\beta }\left(z_{2}\right)+{\left(\frac{1}{2}\right)}^{1-\frac{\beta }{2}}{\mathrm{sig}}^{1-\beta }\left(z_{2}\right)\right]\\ & +\hat{\theta }g_{1}∥S(X)∥\tanh\left(\frac{z_{2}∥S(X)∥}{\rho }\right) \end{array}$$
(32)$$\begin{array}{c} \dot{\hat{\theta }}=rz_{2}g_{1}∥S(X)∥\tanh\left(\frac{z_{2}∥S(X)∥}{\rho }\right)-\varsigma \hat{\theta }-c{\hat{\theta }}^{1+\beta } \end{array}$$
where $\varepsilon_{2}>0$ represents a constant, $\varsigma ={\left(\frac{\pi }{\beta T_{c}}\right)}^{\frac{2}{2-\beta }},c=\frac{\pi (2+\beta )}{2\beta T_{c}r^{\frac{\beta }{2}}\left(1+\beta \right)}$. Invoking Lemma 6 and defining $\omega \left(t\right)=z_{2}∥S(X)∥\tanh\left(\frac{z_{2}∥S(X)∥}{\rho }\right)$, the analysis of the tanh function’s graph indicates that if $z_{2}\geq 0$, then $z_{2}∥S(X)∥\geq 0$ leads to $\tanh\left(\frac{z_{2}∥S(X)∥}{\rho }\right)\geq 0$, denoted as ω(t)≥0; if $z_{2}<0$, then $z_{2}∥S(X)∥\leq 0$ leads to $\tanh\left(\frac{z_{2}∥S(X)∥}{\rho }\right)\leq 0$, denoted as ω(t)≥0. Thus, ω(t) is determined to be a non-negative function. Moreover, selecting $\hat{\theta }\left(0\right)\geq 0$ ensures that for any t≥0, $\hat{\theta }\left(t\right)\geq 0$ invariably holds.

**Remark** **3.**
*Employing the analytical approach outlined in Remark 2, it can be established that ${\dot{\alpha }}_{2}$ is bounded. Moreover, this methodology is similarly utilized to demonstrate that the first derivative of the continuous controller (Equation 33) is bounded.*


#### 2.4.2. Event-Triggered Mechanism Design

In conventional approaches, thresholds for event-triggered strategies are typically constants [36,37]. To enhance communication efficiency and control accuracy in practical applications, time-varying thresholds are indispensable. Therefore, this section introduces an event-triggered mechanism utilizing time-varying thresholds. By employing this mechanism along with the adaptive predefined-time backstepping control strategy described in Section 2.4.1, the proposed controller ensures system stability while avoiding the Zeno phenomenon.

The event-triggered controller has been designed as follows:(33)$$\begin{array}{cc} \psi (t)= & -\left(1+\nu \right)\alpha_{2}\tanh\left(\frac{z_{2}g_{1}\alpha_{2}}{\zeta }\right)\\ & -\left(1+\nu \right){\bar{r}}_{1}\tanh\left(\frac{z_{2}g_{1}{\bar{r}}_{1}}{\zeta }\right) \end{array}$$

The event-triggered mechanism is designed as follows: (34)$$\begin{array}{cc} u(t) & =\psi \left(t_{k}\right),\forall t\in \left[t_{k},t_{k+1}\right). \end{array}$$(35)$$\begin{array}{cc} t_{k+1} & =\inf\left\{t\in R|\left|e(t)\right|\geq \nu \left|u(t)\right|+r_{1}\right\} \end{array}$$
where e(t)=ψ(t)−u(t) denotes the measurement error, and $\zeta ,r_{1},{\bar{r}}_{1}$ represent positive parameters, with the constraints ${\bar{r}}_{1}>\frac{r_{1}}{1-\nu }$ and 0<ν<1. Upon meeting the trigger condition (Equation 35), the current time is set to $t_{k+1}$, and then the updated control signal $\psi (t_{k+1})$ is implemented in the system. The time $t_{k}$ signifies the moment an event is triggered, while $t_{k+1}$ indicates the time at which the controller is updated.

As indicated by (Equation 34) and (Equation 35), there exists a continuous time-varying coefficient $\tau_{2}\left(t\right)$, fulfilling conditions $\tau_{2}\left(t_{k}\right)=0,\tau_{2}\left(t_{k+1}\right)=\pm 1$ and $\left|\tau_{2}\left(t\right)\right|\leq 1$, leads to $\forall t\in \left[t_{k},t_{k+1}\right)$:(36)$$\begin{array}{cc} \psi (t) & =u\left(t\right)+\tau_{2}\left(t\right)\left[\nu \left|u(t)\right|+r_{1}\right]\\ \; & =\left[1+\tau_{1}\left(t\right)\nu \right]u\left(t\right)+\tau_{2}\left(t\right)r_{1} \end{array}$$
where $\tau_{1}\left(t\right)=\tau_{2}\left(t\right)\mathrm{sgn}\left[u\left(t\right)\right]$. Given $\left|\tau_{2}\left(t\right)\right|\leq 1$ and $\left|\mathrm{sgn}[u(t)]\right|\leq 1$ it follows that $\left|\tau_{1}\left(t\right)\right|=\left|\tau_{2}\left(t\right)\mathrm{sgn}\left[u\left(t\right)\right]\right|\leq 1$.

Consequently, we derive
(37)$$\begin{array}{c} u\left(t\right)=\frac{\psi \left(t\right)-\tau_{2}\left(t\right)r_{1}}{1+\tau_{1}\left(t\right)\nu } \end{array}$$

By combining (Equation 33) and (Equation 37), $z_{2}g_{1}u$ can be represented as follows:(38)$$\begin{array}{cc} z_{2}g_{1}u= & \frac{z_{2}g_{1}\psi \left(t\right)-z_{2}g_{1}\tau_{2}\left(t\right)r_{1}}{1+\tau_{1}\left(t\right)\nu }\\ \leq & -\frac{z_{2}g_{1}\left(1+\nu \right)\alpha_{2}\tanh\left(\frac{z_{2}g_{1}\alpha_{2}}{\zeta }\right)}{1+\tau_{1}\left(t\right)\nu }\\ & -\frac{z_{2}g_{1}\left(1+\nu \right){\bar{r}}_{1}\tanh\left(\frac{z_{2}g_{1}{\bar{r}}_{1}}{\zeta }\right)}{1+\tau_{1}\left(t\right)\nu }+\left|\frac{z_{2}g_{1}\tau_{2}\left(t\right)r_{1}}{1+\tau_{1}\left(t\right)\nu }\right| \end{array}$$

The inequality $-\alpha \tanh\left(\frac{\alpha }{\lambda }\right)\leq 0$ invariably holds, and $\frac{1}{1+\nu }\leq \frac{1}{1+\tau_{1}\left(t\right)\nu }$; we obtain
(39)$$\begin{array}{cc} -\frac{z_{2}g_{1}\left(1+\nu \right)\alpha_{2}\tanh\left(\frac{z_{2}g_{1}\alpha_{2}}{\zeta }\right)}{1+\tau_{1}\left(t\right)\nu }\; & \leq -\frac{z_{2}g_{1}\left(1+\nu \right)\alpha_{2}\tanh\left(\frac{z_{2}g_{1}\alpha_{2}}{\zeta }\right)}{1+\nu }\\ & =-z_{2}g_{1}\alpha_{2}\tanh\left(\frac{z_{2}g_{1}\alpha_{2}}{\zeta }\right) \end{array}$$
(40)$$\begin{array}{cc} -\frac{z_{2}g_{1}\left(1+\nu \right){\bar{r}}_{1}\tanh\left(\frac{z_{2}g_{1}{\bar{r}}_{1}}{\zeta }\right)}{1+\tau_{1}\left(t\right)\nu }\; & \leq -\frac{z_{2}g_{1}\left(1+\nu \right){\bar{r}}_{1}\tanh\left(\frac{z_{2}g_{1}{\bar{r}}_{1}}{\zeta }\right)}{1+\nu }\\ & =-z_{2}g_{1}{\bar{r}}_{1}\tanh\left(\frac{z_{2}g_{1}{\bar{r}}_{1}}{\zeta }\right) \end{array}$$

Rearranging the last term in (Equation 38), we obtain:(41)$$\begin{array}{cc} \left|\frac{z_{2}g_{1}\tau_{2}\left(t\right)r_{1}}{1+\tau_{1}\left(t\right)\nu }\right| & \leq \left|z_{2}g_{1}\right|\frac{r_{1}}{1-\nu }\leq \left|z_{2}g_{1}{\bar{r}}_{1}\right| \end{array}$$

By substituting (Equation 39)–(Equation 41) into (Equation 38), we derive the following inequality:(42)$$\begin{array}{cc} z_{2}g_{1}u\leq & -z_{2}g_{1}\alpha_{2}\tanh\left(\frac{z_{2}g_{1}\alpha_{2}}{\zeta }\right)-z_{2}g_{1}{\bar{r}}_{1}\tanh\left(\frac{z_{2}g_{1}{\bar{r}}_{1}}{\zeta }\right)\\ & +\left|z_{2}g_{1}{\bar{r}}_{1}\right|\\ \leq & \left|z_{2}g_{1}\alpha_{2}\right|-z_{2}g_{1}\alpha_{2}\tanh\left(\frac{z_{2}g_{1}\alpha_{2}}{\zeta }\right)+\left|z_{2}g_{1}{\bar{r}}_{1}\right|\\ & -z_{2}g_{1}{\bar{r}}_{1}\tanh\left(\frac{z_{2}g_{1}{\bar{r}}_{1}}{\zeta }\right)+\left|z_{2}g_{1}{\bar{r}}_{1}\right|-\left|z_{2}g_{1}\alpha_{2}\right|\\ & -\left|z_{2}g_{1}{\bar{r}}_{1}\right| \end{array}$$

According to Lemma 1, we obtain
(43)$$\begin{array}{cc} z_{2}g_{1}u & \leq 0.557\zeta +\left|z_{2}g_{1}{\bar{r}}_{1}\right|-\left|z_{2}g_{1}{\bar{r}}_{1}\right|-\left|z_{2}g_{1}\alpha_{2}\right|\\ \; & \leq 0.557\zeta +z_{2}g_{1}\alpha_{2} \end{array}$$

By integrating Lemma 5 with (Equation 30) and (Equation 31), we derive
(44)$$\begin{array}{cc} z_{2}g_{1}\alpha_{2}= & \frac{-z_{2}^{2}{\overset{ˇ}{\alpha_{2}}}^{2}}{\sqrt{z_{2}^{2}{\overset{ˇ}{\alpha_{2}}}^{2}+\varepsilon_{2}^{2}}}\\ < & \varepsilon_{2}-z_{2}\overset{ˇ}{\alpha_{2}}\\ < & \varepsilon_{2}-\frac{\pi }{\beta T_{c}}2^{\beta }{\left(\frac{z_{2}^{2}}{2}\right)}^{1+\frac{\beta }{2}}-\frac{\pi }{\beta T_{c}}{\left(\frac{z_{2}^{2}}{2}\right)}^{1-\frac{\beta }{2}}\\ & -z_{2}\hat{\theta }g_{1}∥S(X)∥\tanh\left(\frac{z_{2}∥S(X)∥}{\rho }\right) \end{array}$$

Finally, substituting (Equation 44) into (Equation 43) yields
(45)$$\begin{array}{cc} z_{2}g_{1}u\leq & -\frac{\pi }{\beta T_{c}}2^{\beta }{\left(\frac{z_{2}^{2}}{2}\right)}^{1+\frac{\beta }{2}}-\frac{\pi }{\beta T_{c}}{\left(\frac{z_{2}^{2}}{2}\right)}^{1-\frac{\beta }{2}}\\ & -z_{2}\hat{\theta }g_{1}∥S(X)∥\tanh\left(\frac{z_{2}∥S(X)∥}{\rho }\right)\\ & +\varepsilon_{2}+0.557\zeta \end{array}$$

Based on the preceding analysis, substituting (Equation 45) into (Equation 29) and rearranging yields
(46)$$\begin{array}{cc} {\dot{V}}_{2}\leq & -\frac{\pi }{\beta T_{c}}{\left(\frac{z_{1}^{2}}{2}\right)}^{1-\frac{\beta }{2}}-\frac{\pi }{\beta T_{c}}2^{\beta }{\left(\frac{z_{1}^{2}}{2}\right)}^{1+\frac{\beta }{2}}-\frac{\pi }{\beta T_{c}}2^{\beta }{\left(\frac{z_{2}^{2}}{2}\right)}^{1+\frac{\beta }{2}}-\frac{\pi }{\beta T_{c}}{\left(\frac{z_{2}^{2}}{2}\right)}^{1-\frac{\beta }{2}}\\ & -z_{2}\hat{\theta }g_{1}∥S(X)∥\tanh\left(\frac{z_{2}∥S(X)∥}{\rho }\right)+z_{2}\theta g_{1}∥S(X)∥\tanh\left(\frac{z_{2}∥S(X)∥}{\rho }\right)-\frac{1}{r}\tilde{\theta }\dot{\hat{\theta }}\\ & +\varepsilon_{2}+\delta \theta g_{1}\rho +\frac{1}{2}\varepsilon^{2}+0.557\zeta +\varepsilon_{1}\\ = & -\frac{\pi }{\beta T_{c}}\left[{\left(\frac{z_{1}^{2}}{2}\right)}^{1-\frac{\beta }{2}}+{\left(\frac{z_{2}^{2}}{2}\right)}^{1-\frac{\beta }{2}}\right]-\frac{\pi }{\beta T_{c}}2^{\beta }\left[{\left(\frac{z_{1}^{2}}{2}\right)}^{1+\frac{\beta }{2}}+{\left(\frac{z_{2}^{2}}{2}\right)}^{1+\frac{\beta }{2}}\right]+\sigma_{1}\\ & +\frac{\tilde{\theta }}{r}\left[rz_{2}g_{1}∥S(X)∥\tanh\left(\frac{z_{2}∥S(X)∥}{\rho }\right)-\dot{\hat{\theta }}\right] \end{array}$$
where $\sigma_{1}=\varepsilon_{2}+\delta \theta g_{1}\rho +\frac{1}{2}\varepsilon^{2}+0.557\zeta +\varepsilon_{1}$ denotes a positive constant.

By substituting adaptive law (Equation 32) into (Equation 46) and simplifying, we derive
(47)$$\begin{array}{cc} {\dot{V}}_{2}\leq & -\frac{\pi }{\beta T_{c}}\left[{\left(\frac{z_{1}^{2}}{2}\right)}^{1-\frac{\beta }{2}}+{\left(\frac{z_{2}^{2}}{2}\right)}^{1-\frac{\beta }{2}}\right]-\frac{\pi }{\beta T_{c}}2^{\beta }[{\left(\frac{z_{1}^{2}}{2}\right)}^{1+\frac{\beta }{2}}\\ & +{\left(\frac{z_{2}^{2}}{2}\right)}^{1+\frac{\beta }{2}}]+\frac{\tilde{\theta }}{r}[rz_{2}g_{1}∥S(X)∥\tanh\left(\frac{z_{2}∥S(X)∥}{\rho }\right)\\ & -rz_{2}g_{1}∥S(X)∥\tanh\left(\frac{z_{2}∥S(X)∥}{\rho }\right)+\varsigma \hat{\theta }+c{\hat{\theta }}^{1+\beta }]\\ & +\sigma_{1}\\ = & -\frac{\pi }{\beta T_{c}}\left[{\left(\frac{z_{1}^{2}}{2}\right)}^{1-\frac{\beta }{2}}+{\left(\frac{z_{2}^{2}}{2}\right)}^{1-\frac{\beta }{2}}\right]-\frac{\pi }{\beta T_{c}}2^{\beta }[{\left(\frac{z_{1}^{2}}{2}\right)}^{1+\frac{\beta }{2}}\\ & +{\left(\frac{z_{2}^{2}}{2}\right)}^{1+\frac{\beta }{2}}]+\frac{\varsigma \tilde{\theta }\hat{\theta }}{r}+\frac{c}{r}\tilde{\theta }{\hat{\theta }}^{1+\beta }+\sigma_{1} \end{array}$$

#### 2.4.3. Stability Analysis

In this section, we will utilize the Lyapunov function, in conjunction with Section 2.4.1 and Section 2.4.2 and relevant mathematical lemmas, to theoretically demonstrate the predefined-time stability of the system.

**Theorem** **1.**
*Considering the robotic manipulator system (Equation 2), virtual controllers (Equation 15) and (Equation 30), actual controller (Equation 37), and the adaptive law (Equation 32), under the event-triggered mechanisms (Equation 34) and (Equation 35) and Lemma 4, the system is PPTS with the system’s error signals $\varkappa ={\left[z_{1},z_{2},\tilde{\theta }\right]}^{T}$ converging within $T_{P}$ to a compact set, where $T_{P}$ denotes the settling time, satisfying $T_{P}<T_{\max}=\sqrt{2}T_{c}$, with all signals in the closed-loop robotic manipulator system being bounded.*


**Proof.** According to (Equation 47), we obtain
(48)$$\begin{array}{cc} {\dot{V}}_{2}\leq & -\frac{\pi }{\beta T_{c}}\left[{\left(\frac{z_{1}^{2}}{2}\right)}^{1-\frac{\beta }{2}}+{\left(\frac{z_{2}^{2}}{2}\right)}^{1-\frac{\beta }{2}}\right]-\frac{\pi }{\beta T_{c}}2^{\beta }[{\left(\frac{z_{1}^{2}}{2}\right)}^{1+\frac{\beta }{2}}\\ & +{\left(\frac{z_{2}^{2}}{2}\right)}^{1+\frac{\beta }{2}}]+\frac{\varsigma \tilde{\theta }\hat{\theta }}{r}+\frac{c}{r}\tilde{\theta }{\hat{\theta }}^{1+\beta }+\sigma_{1} \end{array}$$Utilizing Young’s inequality, we derive
(49)$$\begin{array}{c} \frac{\varsigma \tilde{\theta }\hat{\theta }}{r}\leq -\frac{\varsigma {\tilde{\theta }}^{2}}{2r}+\frac{\varsigma \theta^{2}}{2r} \end{array}$$Applying Lemma 3, we obtain
(50)$$\begin{array}{c} \frac{c}{r}\tilde{\theta }{\hat{\theta }}^{1+\beta }\leq \frac{c}{r}\frac{1+\beta }{2+\beta }\theta^{2+\beta }-\frac{c}{r}\frac{1+\beta }{2+\beta }{\tilde{\theta }}^{2+\beta } \end{array}$$Substituting (Equation 49) and (Equation 50) into (Equation 48) yields
(51)$$\begin{array}{cc} {\dot{V}}_{2}\leq & -\frac{\pi }{\beta T_{c}}\left[{\left(\frac{z_{1}^{2}}{2}\right)}^{1-\frac{\beta }{2}}+{\left(\frac{z_{2}^{2}}{2}\right)}^{1-\frac{\beta }{2}}\right]-\frac{\pi }{\beta T_{c}}2^{\beta }\left[{\left(\frac{z_{1}^{2}}{2}\right)}^{1+\frac{\beta }{2}}+{\left(\frac{z_{2}^{2}}{2}\right)}^{1+\frac{\beta }{2}}\right]\\ & -\frac{\varsigma {\tilde{\theta }}^{2}}{2r}+\frac{\varsigma \theta^{2}}{2r}+\frac{c}{r}\frac{1+\beta }{2+\beta }\theta^{2+\beta }-\frac{c}{r}\frac{1+\beta }{2+\beta }{\tilde{\theta }}^{2+\beta }+\sigma_{1}-{\left(\frac{\varsigma {\tilde{\theta }}^{2}}{2r}\right)}^{1-\frac{\beta }{2}}\\ & +{\left(\frac{\varsigma {\tilde{\theta }}^{2}}{2r}\right)}^{1-\frac{\beta }{2}} \end{array}$$Applying Lemma 7, we derive the following inequality:
(52)$$\begin{array}{c} {\left(\frac{\varsigma {\tilde{\theta }}^{2}}{2r}\right)}^{1-\frac{\beta }{2}}\leq \frac{\varsigma {\tilde{\theta }}^{2}}{2r}+\frac{\beta }{2}{\left(1-\frac{\beta }{2}\right)}^{\frac{2-\beta }{\beta }} \end{array}$$Deriving from Equation (Equation 32), we obtain
(53)$$\begin{array}{c} \varsigma^{1-\frac{\beta }{2}}={\left(\frac{\pi }{\beta T_{c}}\right)}^{\frac{2}{2-\beta }\frac{2-\beta }{2}}=\frac{\pi }{\beta T_{c}} \end{array}$$
(54)$$\begin{array}{cc} \frac{c}{r}\frac{1+\beta }{2+\beta }{\tilde{\theta }}^{2+\beta }= & \frac{c2^{1+\frac{\beta }{2}}r^{\frac{\beta }{2}}\left(1+\beta \right)}{2+\beta }{\left(\frac{{\tilde{\theta }}^{2}}{2r}\right)}^{1+\frac{\beta }{2}}\\ = & \frac{\pi }{\beta T_{c}}2^{\frac{\beta }{2}}{\left(\frac{{\tilde{\theta }}^{2}}{2r}\right)}^{1+\frac{\beta }{2}} \end{array}$$Substituting (Equation 52)–(Equation 54) into (Equation 51) and rearranging yields
(55)$$\begin{array}{cc} {\dot{V}}_{2}\leq & -\frac{\pi }{\beta T_{c}}\left[{\left(\frac{z_{1}^{2}}{2}\right)}^{1-\frac{\beta }{2}}+{\left(\frac{z_{2}^{2}}{2}\right)}^{1-\frac{\beta }{2}}\right]-\frac{\pi }{\beta T_{c}}2^{\beta }\left[{\left(\frac{z_{1}^{2}}{2}\right)}^{1+\frac{\beta }{2}}+{\left(\frac{z_{2}^{2}}{2}\right)}^{1+\frac{\beta }{2}}\right]\\ & -\frac{\varsigma {\tilde{\theta }}^{2}}{2r}+\frac{\varsigma \theta^{2}}{2r}+\frac{c}{r}\frac{1+\beta }{2+\beta }\theta^{2+\beta }-\frac{\pi }{\beta T_{c}}2^{\frac{\beta }{2}}{\left(\frac{{\tilde{\theta }}^{2}}{2r}\right)}^{1+\frac{\beta }{2}}+\sigma_{1}-\frac{\pi }{\beta T_{c}}{\left(\frac{{\tilde{\theta }}^{2}}{2r}\right)}^{1-\frac{\beta }{2}}\\ & +\frac{\varsigma {\tilde{\theta }}^{2}}{2r}+\frac{\beta }{2}{\left(1-\frac{\beta }{2}\right)}^{\frac{2-\beta }{\beta }}\\ = & -\frac{\pi }{\beta T_{c}}\left[{\left(\frac{z_{1}^{2}}{2}\right)}^{1-\frac{\beta }{2}}+{\left(\frac{z_{2}^{2}}{2}\right)}^{1-\frac{\beta }{2}}\right]-\frac{\pi }{\beta T_{c}}2^{\beta }\left[{\left(\frac{z_{1}^{2}}{2}\right)}^{1+\frac{\beta }{2}}+{\left(\frac{z_{2}^{2}}{2}\right)}^{1+\frac{\beta }{2}}\right]\\ & -\frac{\pi }{\beta T_{c}}2^{\frac{\beta }{2}}{\left(\frac{{\tilde{\theta }}^{2}}{2r}\right)}^{1+\frac{\beta }{2}}-\frac{\pi }{\beta T_{c}}{\left(\frac{{\tilde{\theta }}^{2}}{2r}\right)}^{1-\frac{\beta }{2}}+\sigma_{2} \end{array}$$
where $\sigma_{2}=\sigma_{1}+\frac{\beta }{2}{\left(1-\frac{\beta }{2}\right)}^{\frac{2-\beta }{\beta }}+\frac{\varsigma \theta^{2}}{2r}+\frac{c}{r}\frac{1+\beta }{2+\beta }\theta^{2+\beta }$, $\sigma_{2}>0$.Applying Lemma 2 yields the following inequalities:
(56)$$\begin{array}{cc} {\left(\frac{z_{1}^{2}}{2}\right)}^{1-\frac{\beta }{2}}+{\left(\frac{z_{2}^{2}}{2}\right)}^{1-\frac{\beta }{2}} & \geq {\left(\frac{z_{1}^{2}}{2}+\frac{z_{2}^{2}}{2}\right)}^{1-\frac{\beta }{2}} \end{array}$$
(57)$$\begin{array}{cc} {\left(\frac{z_{1}^{2}}{2}\right)}^{1+\frac{\beta }{2}}+{\left(\frac{z_{2}^{2}}{2}\right)}^{1+\frac{\beta }{2}} & \geq 2^{-\frac{\beta }{2}}{\left(\frac{z_{1}^{2}}{2}+\frac{z_{2}^{2}}{2}\right)}^{1+\frac{\beta }{2}} \end{array}$$By combining inequalities, (Equation 56), (Equation 57) and (Equation 55) can be reformulated as
(58)$$\begin{array}{cc} {\dot{V}}_{2}\leq & -\frac{\pi }{\beta T_{c}}{\left(\frac{z_{1}^{2}}{2}+\frac{z_{2}^{2}}{2}\right)}^{1-\frac{\beta }{2}}-\frac{\pi }{\beta T_{c}}2^{\frac{\beta }{2}}{\left(\frac{z_{1}^{2}}{2}+\frac{z_{2}^{2}}{2}\right)}^{1+\frac{\beta }{2}}\\ & -\frac{\pi }{\beta T_{c}}2^{\frac{\beta }{2}}{\left(\frac{{\tilde{\theta }}^{2}}{2r}\right)}^{1+\frac{\beta }{2}}-\frac{\pi }{\beta T_{c}}{\left(\frac{{\tilde{\theta }}^{2}}{2r}\right)}^{1-\frac{\beta }{2}}+\sigma_{2}\\ = & -\frac{\pi }{\beta T_{c}}\left[{\left(\frac{z_{1}^{2}}{2}+\frac{z_{2}^{2}}{2}\right)}^{1-\frac{\beta }{2}}+{\left(\frac{{\tilde{\theta }}^{2}}{2r}\right)}^{1-\frac{\beta }{2}}\right]\\ & -\frac{\pi }{\beta T_{c}}2^{\frac{\beta }{2}}\left[{\left(\frac{z_{1}^{2}}{2}+\frac{z_{2}^{2}}{2}\right)}^{1+\frac{\beta }{2}}+{\left(\frac{{\tilde{\theta }}^{2}}{2r}\right)}^{1+\frac{\beta }{2}}\right]+\sigma_{2} \end{array}$$Reapplying Lemma 2 yields the following inequalities:
(59)$$\begin{array}{cc} {\left(\frac{z_{1}^{2}}{2}+\frac{z_{2}^{2}}{2}\right)}^{1-\frac{\beta }{2}}+{\left(\frac{{\tilde{\theta }}^{2}}{2r}\right)}^{1-\frac{\beta }{2}} & \geq {\left(\frac{z_{1}^{2}}{2}+\frac{z_{2}^{2}}{2}+\frac{{\tilde{\theta }}^{2}}{2r}\right)}^{1-\frac{\beta }{2}} \end{array}$$
(60)$$\begin{array}{cc} {\left(\frac{z_{1}^{2}}{2}+\frac{z_{2}^{2}}{2}\right)}^{1+\frac{\beta }{2}}+{\left(\frac{{\tilde{\theta }}^{2}}{2r}\right)}^{1+\frac{\beta }{2}} & \geq 2^{-\frac{\beta }{2}}{\left(\frac{z_{1}^{2}}{2}+\frac{z_{2}^{2}}{2}+\frac{{\tilde{\theta }}^{2}}{2r}\right)}^{1+\frac{\beta }{2}} \end{array}$$Integrating inequalities (Equation 59) and (Equation 60) allows (Equation 58) to be reformulated as
(61)$$\begin{array}{c} {\dot{V}}_{2}\leq -\frac{\pi }{\beta T_{c}}\left(V_{2}^{1-\frac{\beta }{2}}+V_{2}^{1+\frac{\beta }{2}}\right)+\sigma_{2} \end{array}$$
where $\sigma_{2}=\varepsilon_{1}+\varepsilon_{2}+\delta \theta g_{1}\rho +\frac{1}{2}\varepsilon^{2}+0.557\zeta +\frac{\beta }{2}{\left(1-\frac{\beta }{2}\right)}^{\frac{2-\beta }{\beta }}+\frac{\varsigma \theta^{2}}{2r}+\frac{c}{r}\frac{1+\beta }{2+\beta }\theta^{2+\beta }$ is regarded as a bounded positive constant.Following Lemma 4 and (Equation 61), the system is PPTS, and the error signals $\varkappa ={\left[z_{1},z_{2},\tilde{\theta }\right]}^{T}$ are capable of converging to the compact set:
(62)$$\Delta =\left\{\lim_{t\to T_{P}}\varkappa |V_{2}\leq \min\left\{{\left(\frac{2\beta T_{c}\sigma_{2}}{\pi }\right)}^{\frac{2}{2-\beta }},{\left(\frac{2\beta T_{c}\sigma_{2}}{\pi }\right)}^{\frac{2}{2+\beta }}\right\}\right\}$$
where $T_{P}$ denotes the settling time and meets the condition $T_{P}<T_{\max}=\sqrt{2}T_{c}$, clearly indicating that $z_{1},z_{2},\tilde{\theta }$ are bounded. Given that θ is a constant and assuming conditions for $\theta =\hat{\theta }+\tilde{\theta }$, the boundedness of $\hat{\theta }$ is guaranteed. Furthermore, given that $\hat{\theta },z_{1},z_{2},{\dot{y}}_{d}$ are bounded, and under the conditions of $\left|\tanh(\cdot )\right|<1$, it follows that $\alpha_{1},\alpha_{2}$ are also bounded. Further, from Equations (Equation 33) and (Equation 37), it is derived that controllers ψ(t),u(t) are bounded. Moreover, given Equation (Equation 14) and the boundedness of $y_{d}$, it follows that $x_{1}$ and $x_{2}$ are bounded. Consequently, all signals within the closed-loop robotic manipulator system are bounded, concluding the proof of Theorem 1. □

**Remark** **4.**
*Based on the stability analysis, by adjusting the predefined parameter $T_{c}$, we can establish an upper bound for the settling time of the closed-loop system at $T_{P}<\sqrt{2}T_{c}$. Additionally, decreasing $T_{c}$ results in an accelerated convergence rate.*


### 2.5. Feasibility Analysis

Subsequently, the proposed event-triggered mechanism’s feasibility is substantiated by the exclusion of the Zeno phenomenon.

**Theorem** **2.**
*Given the robotic manipulator system (Equation 2), alongside the virtual controllers (Equation 15) and (Equation 30), the event-triggered controller (Equation 33) and adaptive law (Equation 32), and event-triggered mechanisms (Equation 34) and (Equation 35), a positive constant $t^{*}$ exists, ensuring $\forall k\in Z^{*},t_{k+1}-t_{k}\geq t^{*}$ and thus eliminating the Zeno phenomenon.*


**Proof.** Given $\forall t\in \left[t_{k},t_{k+1}\right)$, and with $u\left(t\right)=\psi \left(t_{k}\right)$ as a constant, coupled with e(t)=ψ(t)−u(t), it can be deduced that $\dot{e}\left(t\right)=\dot{\psi }\left(t\right)$. Further analysis yields that $\frac{d}{dt}\left|e(t)\right|=\frac{d}{dt}{\left(e\times e\right)}^{\frac{1}{2}}=\mathrm{sgn}(e)\dot{e}\leq \left|\dot{\psi }\right|$ holds true. Following the preceding analysis and Remark 3, we conclude that the first derivative of ψ(t) is bounded. Consequently, a positive constant *s* exists, ensuring that $\left|\dot{\psi }\left(t\right)\right|\leq s$. With $e(t_{k})=0$, alongside $\lim_{t\to t_{k+1}}\left|e(t)\right|\geq \nu \left|u(t)\right|+r_{1}$ and $\frac{d}{dt}\left|e(t)\right|\leq \left|\dot{\psi }\right|\leq s$, it can be inferred that $\frac{\lim_{t\to t_{k+1}\left|e(t)\right|-\left|e(t_{k})\right|}}{t_{k+1}-t_{k}}=\frac{\lim_{t\to t_{k+1}}\left|e(t)\right|}{t_{k+1}-t_{k}}\leq s$. Further deduction reveals that $t_{k+1}-t_{k}\geq \frac{\nu \left|u(t)\right|+r_{1}}{s}$. Hence, a positive constant $t^{*}$ necessarily exists, ensuring $t_{k+1}-t_{k}\geq t^{*}$, which in turn circumvents the Zeno phenomenon. □

## 3. Results

In this section, the effectiveness of the proposed control strategy is verified through simulation for a single-arm robotic system characterized by uncertainty. The dynamic model of the single-arm robot is presented in (Equation 1), with the arm parameters and the target reference signal given in Table 1. The system states $x_{1},x_{2}$ are initially set to [1,−1.5], with the adaptive parameter initialized at $\hat{\theta }\left(0\right)=1$. The RBFNN comprises seven nodes, with Gaussian function centers μ uniformly distributed across the interval [−1.5,1.5], and the Gaussian function width is η=10. The design parameters for the virtual controllers (Equation 15) and (Equation 30), actual controller (Equation 37), event-triggered mechanism (Equation 35), and adaptive law (Equation 32) are selected as $\beta =\frac{4}{9},\rho =1,\varepsilon_{1}=0.001,\varepsilon_{2}=0.001,r_{1}=0.5,\nu =0.2$, ${\bar{r}}_{1}=11$, ζ=5,r=0.1. To further ascertain the efficacy of the developed predefined-time schemes, two distinct $T_{c}$ values were chosen, specifically 3 s and 6 s. A simulation duration of 20 s was chosen, using degrees to show the tracking errors, which can clearly represent the tracking performance. The simulation results are presented in Figure 3, Figure 4, Figure 5, Figure 6, Figure 7, Figure 8 and Figure 9.

## 4. Discussion

In this section, we further describe and analyze the simulation results of Section 3. Figure 3 illustrates the trajectory of the tracking error $z_{1}$ for the system’s output signal across various predefined times. It is observed that the tracking error converges to a small vicinity around the origin before the designated upper bound of the settling time $\sqrt{2}T_{c}$, meeting the anticipated control objective. Furthermore, a smaller value of the parameter $T_{c}$ correlates with a faster convergence speed of the tracking error. Figure 4 and Figure 5 depict the trajectories of the ideal signal $y_{d}$ and the actual position signal $x_{1}$, along with the trajectories of the ideal angular speed signal ${\dot{y}}_{d}$ and the actual angular speed signal $x_{2}$, under various predefined time parameters. From Figure 3, Figure 4 and Figure 5, it is evident that when the parameter $T_{c}$ is set to 6 s, the actual stabilization time of the system increases. Additionally, the boundary of the system’s settling time can be clearly determined in advance by specifying the parameter $T_{c}$. This verifies the effectiveness of the proposed predefined-time controller. Figure 6 presents the trajectory of the control signal. This demonstrates the effective role of the event-triggered mechanism. Figure 7 illustrates the trajectory curve of the adaptive parameter $\hat{\theta }$. The analysis in Figure 3, Figure 4, Figure 5, Figure 6 and Figure 7 reveals that the control strategy ensures boundedness of the closed-loop system’s angles, angular velocities, controller outputs, and adaptive parameter signals while also guaranteeing that tracking errors converge to a small neighborhood near the origin within the predefined time frame. Figure 8 displays the time intervals between successive event triggers. Statistical analysis reveals that the trigger counts were 986 and 735 for $T_{c}$ values of 3 s and 6 s, respectively. When $T_{c}$ = 3 s is used as an example, the total simulation time is 20 s. It is known that the sampling frequency of classic robotic arm controllers ranges between 500 and 1000/s. Therefore, employing the event-triggered mechanism with time-varying thresholds can save at least [1−986/(500×20)]×100% = 90.14% of communication resources. Figure 9 displays the event-triggering images for fixed and variable thresholds at $T_{c}$ = 3 s. According to statistics, using a fixed threshold resulted in 1582 event triggers, while using a variable threshold resulted in 986 event triggers. Based on the calculations, using fixed and variable thresholds can save at least 84.18% and 90.14% of communication resources, respectively. Moreover, based on the graphs, it is evident that throughout the entire control process, the fixed threshold event-triggering mechanism wastes more communication resources (more triggers with shorter intervals between them), which leads to larger control errors. This further emphasizes the superiority of our variable threshold-triggering mechanism. In conclusion, the simulation outcomes align with the theoretical predictions, demonstrating that the designed controller effectively achieves the control objectives and meets the system’s performance specifications.

## 5. Conclusions

This paper introduces an event-triggered, adaptive predefined-time tracking control method for a robotic manipulator system characterized by uncertainty. This approach ensures that the tracking error converges to a neighborhood near the origin within a predefined time while allowing for the predetermination of the settling time’s upper bound through the adjustment of a straightforward control parameter, thus favoring compliance with real system demands. Additionally, this method conserves communication resources effectively without compromising system control performance. The efficacy of the proposed control approach was substantiated through simulations. While the control strategy exhibits strong convergence and timely responses, its actual performance in real-world scenarios may be compromised by overlooked factors, such as inadequate system modeling or failures in the robotic manipulators’ actuators, thus failing to meet the anticipated standards. Moreover, future work should focus on developing a more precise mathematical model while addressing additional challenges the robotic manipulator might face in practical applications, such as state error constraints and actuator failures, with the goal of devising an enhanced control strategy.

## Figures and Tables

**Figure 1 sensors-24-04573-f001:**
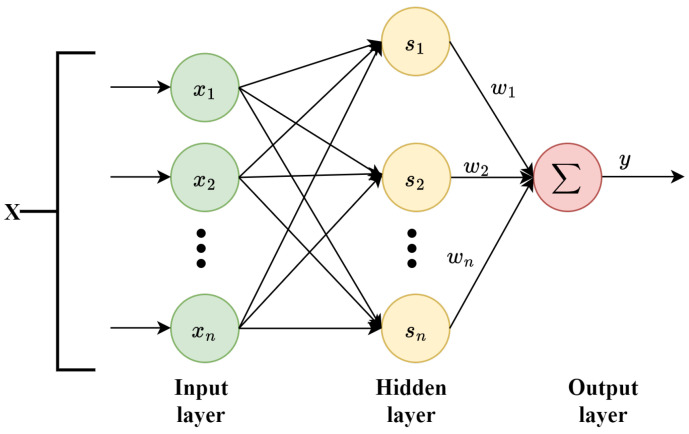
RBF neural network.

**Figure 2 sensors-24-04573-f002:**
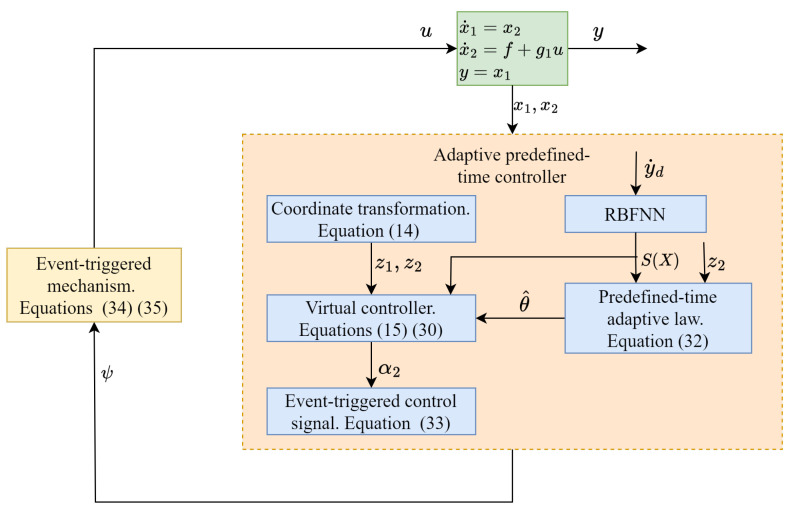
Control block diagram.

**Figure 3 sensors-24-04573-f003:**
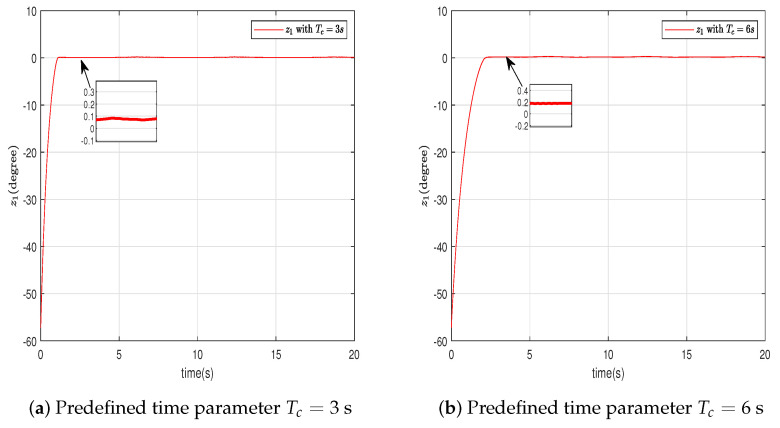
Trajectory of the angular tracking error $z_{1}$.

**Figure 4 sensors-24-04573-f004:**
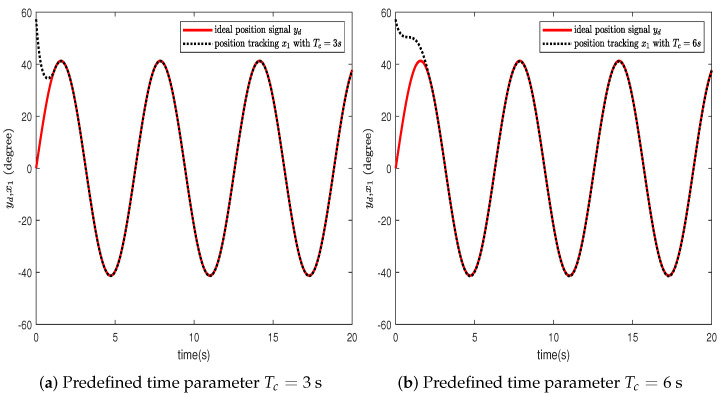
Trajectories of the ideal reference signal $y_{d}$ and the actual system output signal $x_{1}$.

**Figure 5 sensors-24-04573-f005:**
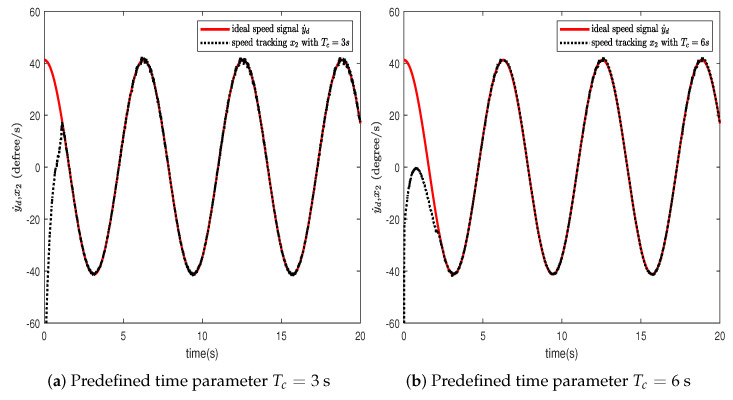
Trajectories of the ideal angular speed signal ${\dot{y}}_{d}$ and the actual angular speed signal $x_{2}$.

**Figure 6 sensors-24-04573-f006:**
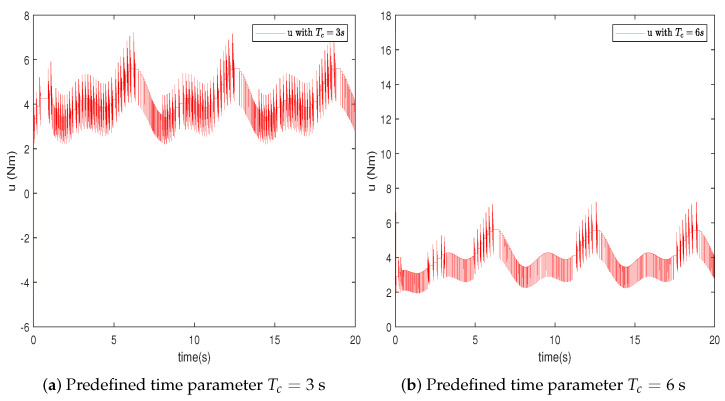
Trajectory of the controller output *u*.

**Figure 7 sensors-24-04573-f007:**
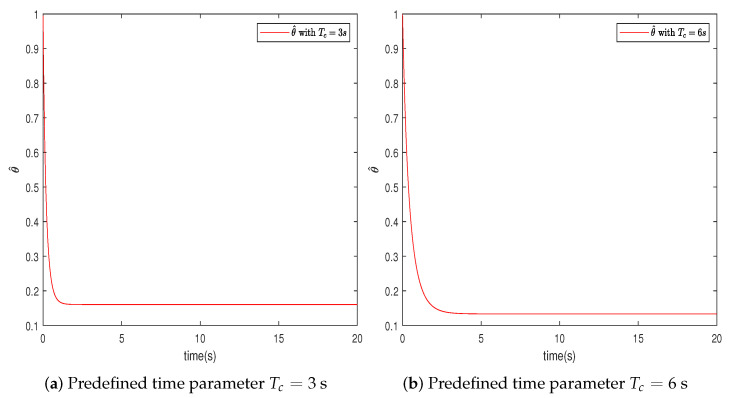
Trajectory of the adaptive parameter $\hat{\theta }$.

**Figure 8 sensors-24-04573-f008:**
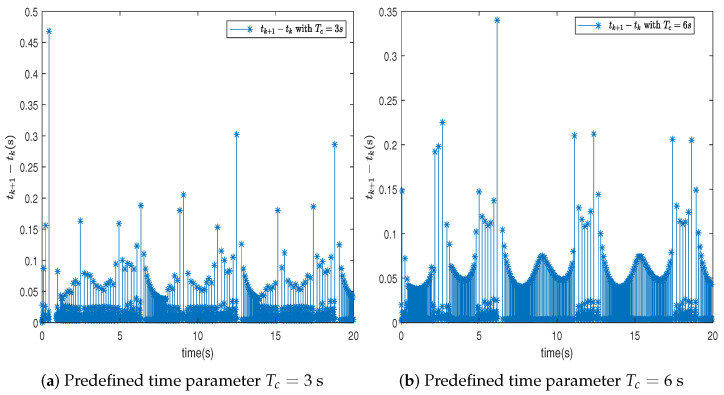
The triggering instants and time intervals $t_{k+1}-t_{k}$.

**Figure 9 sensors-24-04573-f009:**
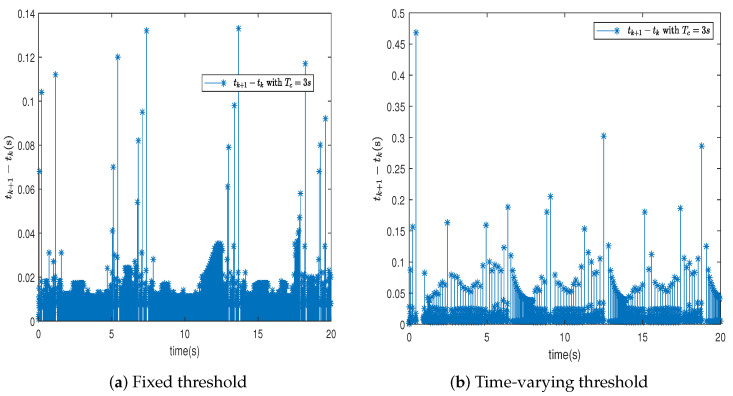
Comparison of adjacent event trigger time interval with fixed threshold and variable threshold with $T_{c}=3s$.

**Table 1 sensors-24-04573-t001:** Model parameters.

Parameter	Value	Units
m	1	kg
l	0.5	m
$c_{0}$	1	N·m·s/rad
g	9.8	m/s^2^
d	sin(t)	N·m
$y_{d}$	0.72sin(t)	rad

## Data Availability

Data sharing is not applicable to this article because no datasets were generated or analyzed during this study.

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
