# Peer review of "Predefined-Time Adaptive Neural Tracking Control for a Single Link Manipulator with an Event-Triggered Mechanism"

_sensors, 2024, doi:10.3390/s24144573_

Round 1
Reviewer 1 Report
Comments and Suggestions for Authors
The manuscript presents a novel adaptive trajectory tracking control method for nonlinear systems to achieve curve reference tracking. The method is rigorously developed and its stability is checked by Lypunov theory. The method also guarantees that tracking errors converge to a small vicinity of the origin within a predefined timeframe. Finally, the efficacy of the control scheme is presented by the simulation results of a nonlinear manipulator system for trajectory tracking.
The paper is difficult to read due to its difficult matter and not proper organization of the text. There are some minor grammatical errors in the text andsome missing information in figures:
1. eq 6: The dot above V(x) must be only above V!
2. Would you explain, what is Tc in the statemant immediately after the eq. 7. What you would like to achieve with changing Tc?
3. Line 355: The figures must be referred as Figs. 1 to 6, or Figures 1 to 6, and noz as Figure.1 to 6. The same mistake continues through the rest of text.
4.What are the units on the y-axes of Figures from 1 to 6? They are all missing.
5. It is extremely difficult to follow the derivation of equations trough the maniscript. Can you divided the equation derivation to more subsection and in the beginning of section describe with the text what would follow and what is the reason to derive those equations. I also recommend you to write also with the text in the end of those newly introduced subsections, what is the result of derivation. This would help the readers easy to understand the equation development. I also recommend to introduce a new figure, where the control scheme would be described precisely with accordance of developed control law u(t).
6. In section 7, you wrote that you conserve the communication resources effectively. Can you describe this sentence with numbers. For example: instead of sampling time "1-2 ms", which is the usually a sampling time for the classical robot controller, you achieved the the average sampling time "15 ms <- I am jus guessing." This kind of information would be a nice information in the discussing the information of figure 6. I also think, that you have to perform more discussion of the results. Overall, the discussion of good results is very poor.
7. The nonlinear plant for your adaptive controller has only viscous friction. THis is not enough, because you are using direct drive robot mechanism (without gear-boxes), so the impact of all frictions in such a mechanism is approximately 20% of all torques (Coulumb friction plus negative viscous friction). This kind of friction, I believe, would make huge problems to your adaptive controller, especially with the velocity of robot axes close to zero. So, I recommend you to include the full model of friction in the model and investigate the velocity during trajectory tracking (actual angular speed) in the vicinity of zero. As, I see you already have problems, when you fast decrease, or increase the speed on the top or on the bottom of curve on Fig. 3.
8. I also highly recommend you to rewrite (reorganize) the complete manuscript, according to the rules of the journal Sensors, where is demanded by the editor to organize the section as: 1. Introduction, 2. Material and Methods (in you case only Methods), 3. Results, 4. Discussion, 5 Conclusion, if needed. (see: Instruction for authors in chapter: Research Manuscript Sections).
9. I understand that your research was the theoretical research, but the ultimate proof would be if this method would work in the real engineering environment on the real direct-drive robot mechanism. This would be really extraordinary scientific achievement.
Comments on the Quality of English Language
The English language is quite good, only some small mistakes have to be improved.
Reviewer 2 Report
Comments and Suggestions for Authors
In this manuscript, the authors design an event-triggered, predefined-time adaptive neural tracking control for single link manipulator. Rigorous stability analysis and simulation results have been provided. It is well-written and logic. Only the following issues need to be dealt with:
1. The block diagram of the used RBFNN should be presented.
2. All parameters used in simulation tests should be summarized in a Table.
Round 2
Reviewer 1 Report
Comments and Suggestions for Authors
The authors have improved the paper according to my suggestions, but there are still 2 major problems, need to be resolved:
1) I suggest you (or better you must do) to put sections 2, 3 in 4 in one section called Methods. The sections 2, 3, 4 in the version 2 of the manuscript renumber to 2.1, 2.2, 2.3 and 2.4, respectively. The section 5. Results must be divided to two sections: Section 3. Results, where only results has to be included, without comments or discussions. Then, you have to make a new section 4. Discussion, where the comments of the results and other type of discussion information would be placed.
2) You told in your comments that the velocity error is small, only 2%. I checked your information and discovered, that for 1 D.O.F robot mechanism you used to control this 2 % of velocity error means the trajectory tracking error in cartesian coordinates for both peaks of velocity (+0.7 rad/s at time 7 s and -0.7 rad/s at time 9.5 s) means approximately 0.014 rad of positional error or in cartesian coordinates 7 mm for the case TC=3 s. The standard for trajectory tracking error, for example industrial welding robot arm is less than 0.15 mm. So, this is quite problematic, because your 1 D.O.F. robot mechanism has mass 1 kg, the length only 0,5 m etc. What do you think what would be the position error in cartesian coordinates if you have mass of few hundred kg and 6 D.O.F robot arm. I believe, it would be much higher than a few cm. I also saw in the figure 8 that the sampling time (tk+1 - tk ) increased up to almost 0.4 s in the times around 7 s and 9,5 s, where is the highest velocity of your 1 D.OF robot mechanism. For me, it would be really strange, if your controller would increase the sampling time in the case of increased velocity. I believe that this increase of sampling time also produce the higher positional error z1. So, please make a discussion about that in your manuscript. By my opinion, the controller doesn't work well. So, try to explain, why this increase of position error happened? Is this correlated with the increase of sampling time? If this is so, than the controller doesn't work well!
